# Development of a Quick-Install Rapid Phenotyping System

**DOI:** 10.3390/s23094253

**Published:** 2023-04-25

**Authors:** Roberto M. Buelvas, Viacheslav I. Adamchuk, John Lan, Valerio Hoyos-Villegas, Arlene Whitmore, Martina V. Stromvik

**Affiliations:** 1Department of Bioresource Engineering, Macdonald Campus, McGill University, 21 111 Lakeshore, Ste-Anne-de-Bellevue, QC H9X 3V9, Canada; 2Department of Plant Science, Macdonald Campus, McGill University, 21 111 Lakeshore, Ste-Anne-de-Bellevue, QC H9X 3V9, Canada

**Keywords:** multispectral sensors, plant phenotyping, sensing system, ultrasonic sensors, vegetation index

## Abstract

In recent years, there has been a growing need for accessible High-Throughput Plant Phenotyping (HTPP) platforms that can take measurements of plant traits in open fields. This paper presents a phenotyping system designed to address this issue by combining ultrasonic and multispectral sensing of the crop canopy with other diverse measurements under varying environmental conditions. The system demonstrates a throughput increase by a factor of 50 when compared to a manual setup, allowing for efficient mapping of crop status across a field with crops grown in rows of any spacing. Tests presented in this paper illustrate the type of experimentation that can be performed with the platform, emphasizing the output from each sensor. The system integration, versatility, and ergonomics are the most significant contributions. The presented system can be used for studying plant responses to different treatments and/or stresses under diverse farming practices in virtually any field environment. It was shown that crop height and several vegetation indices, most of them common indicators of plant physiological status, can be easily paired with corresponding environmental conditions to facilitate data analysis at the fine spatial scale.

## 1. Introduction

A High-Throughput Plant Phenotyping (HTPP) platform is a system used to measure plant traits. Properties commonly measured by such platforms include the architecture of roots, leaves, fruits, and entire canopies; moreover, these platforms may measure processes such as photosynthesis, specific functions of organs or systems, and tolerance to different types of stress [1]. To achieve this, different types of sensors are used, either to obtain direct measurements of the desired traits, or to produce intermediate values that relate to a particular trait. According to [2], some of the most prominent types of sensors used in HTPP are:Infrared thermography and imagery to scan temperature profiles/transpiration;Fluorescent microscopy/spectroscopy to assess photosynthetic rates;3D reconstruction to assess plant growth rate and structure;Light Detection and Ranging (LiDAR) to measure growth rates;Magnetic resonance imaging and positron emission tomography to measure growth patterns, root/leaf physiology, water relations, and/or assimilate translocation properties;Canopy spectral reflectance for monitoring dynamic complex traits;Nuclear magnetic resonance for monitoring the structure of tissues, mapping water movements, and monitoring sucrose allocation;Digital RGB imaging for recording data on various attributes of roots, shoots, leaves, seeds, and grains.

However, in most cases, HTPP platforms rely on sensor fusion to provide more complex information. While this increases the computational costs, generally it is preferred as it provides more insight into the same area. An example of this approach is presented in the Breed-Vision platform [3]. Other institutions and researchers have also developed HTPP platforms, some of which are offered commercially, such as: LemnaTec GmbH (Mention of a trade name, proprietary product, or company name is for presentation clarity and does not imply endorsement by the authors or McGill University, nor the exclusion of other products that may also be suitable) (Aachen, Germany), Phenokey BV (Gravenzande, The Netherlands), PhenoSpex (Heerlen, The Netherlands), Photon System Instruments spol. s r. o. (Drásov, Czechia), Wiwam (Eeklo, Belgium), and We Prove Solutions (De Lier, The Netherlands) [4]. In addition, companies may also offer HTPP as a service, e.g., PhenoFab (KeyGene N.V., Nanjing, China).

As mentioned previously, many implementations exist; nevertheless, HTPP platforms have not achieved their full adoption potential [5]. A major portion of the current offering is focused on controlled environments, such as greenhouses [6]. Furthermore, even though many of these implementations have taken advantage of such techniques as deep learning to process the large amount of data, difficulties remain with modelling, especially given the multiple kinds of sensors and data sources involved in modern phenotyping operations [7]. Finally, the significant costs related to the implementation of phenotyping platforms are a barrier to adoption [8]. Even so phenotyping systems requiring the greatest human/capital investments can provide important cost benefits, especially when genetic gains are improved significantly despite the initial cost. A better understanding of plant physiology and genetics can be developed into novel phenotyping assays that are more accessible, and potentially provide molecular markers for rapid screening of plant traits [9].

Plant phenotyping has four uses: plant breeding, optimizing fertilization, detecting diseases, pests, and weeds, and powering Decision Support Systems (DSS) that provide economic and agronomic advice to farmers [10]. Plant breeding is a key area where HTPP systems could provide benefits [11]. HTPP developments to quantitatively measure key traits will increase accuracy and reduce the costs of the selection process. In combination with genomics, plant phenotyping can be used to find Quantitative Trait Loci (QTL), i.e., measured plant traits can be linked with sections of DNA sequences. This use highlights the importance of HTPP in facilitating plant breeding applications. For example, UAV-based phenotyping platforms have been used to identify QTLs for resistance to bacterial blight in rice [12] and to identify genomic regions related to nitrogen response in maize [13].

Increased data resolution, both in spatial and temporal terms, is the main advantage of HTPP when compared with other phenotyping approaches. Acquiring plant trait information with higher spatial resolution improves the characterization of within-field variability. In turn, shifting the focus to individual plants or individual plant organs, rather than entire canopies, becomes a possibility for agronomists and breeders. Higher temporal resolutions provide for the identification of time-dependent phenomena such as time-interval-specific QTL [14].

Incorporating weather and environmental data in field experiments is important in order to account for the non-genetic factors of measured plant traits. This approach is sometimes referred to as envirotyping. For plant breeding, envirotyping could increase selection accuracy, speed up variety commercialization, and optimize variety evaluation [15]. Environmental conditions also influence the readings of other sensors, such as vegetation indices, even when active sensors are used [16]. Envirotyping aids in the comparison of multi-year experiments and helps with the generalization potential of the experimental results, guaranteeing that new varieties will express their desired characteristics in a variety of locations and climates. The most common types of sensing systems for the purpose of envirotyping rely on fixed structures, either as a weather station placed in a central location, or as Wireless Sensor Networks (WSN) which can cover a wide area. Weather stations generally do not consider the spatial variability of atmospheric conditions at the within-the-field scale. WSN can map environmental factors in that scale effectively, but the installation costs scale up with the area covered as more nodes are required in the network. An alternative that could be more economical is a mobile platform, which collects environmental measurements, and, in turn, this measurement would be complemented by weather station data.

A key component in the design of an HTPP platform is the choice of mounting structure or vehicle. Both outdoor and indoor HTPP platforms may be based on fixed or mobile structures. HTPP platforms with a focus on controlled environments often rely on fixed structures such as conveyor belts or wire harnesses [17]. Whereas HTPP platforms used for outdoor conditions frequently include ground vehicles and UAVs [2]. Handheld setups are used both for indoor and outdoor conditions [18,19,20]. The selection of the platform is also guided by the desired spatial and temporal resolution, and the scale and scope of the experiments to be performed. For example, UAV platforms enable the monitoring of small plots or field scale trials with numerous genotypes, but the satellite platforms are more suited for yield prediction [21]. On the other hand, the higher spatial resolution of UAVs allows the evaluation of intercropping systems, which satellites generally struggle to do [22,23].

Compared to HTTP platforms based on ground vehicles, platforms on UAVs have the advantage of faster measurements without the risk of damaging trial plots. Moreover, UAVs have become more user-friendly [24] and low-cost [11]. Another advantage is that they allow for more uniform coverage [25]. Satellite-based platforms present similar advantages to UAVs, but the revisit frequency and delay between the images being taken and them becoming accessible might be possible additional concerns [13]. Since UAVs typically fly between 10 m and 200 m, and remote sensing satellites are located at an altitude of about 700 km [13], the spatial resolution is not as good as that of proximal sensing with handheld sensors and ground vehicles. The distance between the sensor and its target in a ground vehicle is generally less than 1 m and there may be contact with certain types of sensors. In turn, this can result in better predictions. For example, higher coefficients of determination were achieved in most conditions when predicting durum wheat’s grain yield from VIs collected from the ground, than from an aerial platform in [26]. On the other hand, ground-based vehicles can sustain longer operational times and carry much heavier payloads than UAVs, which allows them to use multiple types of sensors at the same time. Combining different types of measurements tends to provide better results during modelling. However, it is worth noting that different types of measurements can be collected from the same sensor, as evidenced by [27]. Depending on the size of the vehicle and the equipped sensors, platforms based on ground vehicles can also provide sub-canopy information, which UAVs generally cannot [28]. In certain areas, there are also legal restrictions to where UAVs are allowed to be used and how close to the ground or to people they are permitted to fly. Rain and high winds can be more of a problem for a UAV than for a ground vehicle. In summary, all platforms have their advantages and disadvantages.

HTPP platforms based on ground vehicles typically have their sensors mounted in the front or rear of the vehicle. In this scenario, the vehicle drives through, or above, the planted plots. The risk of damaging the soil or the plants with the vehicle, and the possibility for the crop rows’ conditions to limit circulation efficiency, are challenges in the implementation of such HTPP platforms. To overcome these challenges, the vehicle may be equipped with lateral booms reaching over the plots so that it can drive on all-weather traffic lanes adjacent to the plot [29]. In this setup, multiple sets of sensors may be placed along the boom to cover multiple crop rows from a nadir view. The feature of autonomous driving is also related to HTPP platforms based on ground vehicles. Both full and partial autonomy versions of self-driving vehicles show great potential to reduce the workload of operators, while making the entire process faster and more efficient. Many implementations of autonomous navigation for agricultural vehicles leverage local information, such as crop rows, to improve the algorithms of the platform [30].

The goal of this project was to design a HTPP platform for use in field conditions. To illustrate the system’s performance, it was to be deployed in a field planted with dry beans (*Phaseolus vulgaris* L.). The system should be able to take canopy measurements from different types of crops at an early stage, and be based on the quick-install concept [31]. In this way, users can mount the system on a vehicle they already own and unmount it if they need the vehicle for a different operation involving other attachments. The system should be designed to target individual rows or plots, rather than an entire field, in order to take advantage of higher spatial resolution. The system should also integrate envirotyping. The application of HTPP in beans and pulses is not typically found in the literature [32]. The completion of this goal is considered a step towards the ultimate goal of equipping all kinds of growers with a tool for rapid, non-destructive, reliable, and affordable assessment of their crops. It is foreseen that this tool will be helpful for small breeding centers in need of an early signature of hybrid performance.

## 2. Materials and Methods

### 2.1. Benchmark Handheld Setup

Before designing the HTPP platform, a handheld version of the system with a subset of its components was built with the multispectral sensor ACS-430 (Holland Scientific, Inc., Lincoln, NE, USA), the ultrasonic sensor ToughSonic 14 (Senix Corp., Hinesburg, VT, USA), a 6000 mAh power bank, and the tablet Yuma 2 (Trimble Inc., Sunnyvale, CA, USA) with a built-in GNSS receiver. This handheld setup was used to both prototype the operation of the HTPP platform and as a benchmark to compare its ergonomic performance. The basic structure of the setup emulates a staff with sensors near the top. Figure 1 illustrates the handheld setup. The handheld version of the platform ran a preliminary version of the same code, albeit with reduced functionalities.

### 2.2. Electronic Subsystem of the HTPP Platform

The sensors included in the HTPP platform were:6 multispectral sensors ACS-435 (Holland Scientific, Inc., Lincoln, NE, USA);6 ultrasonic sensors ToughSonic 14 (Senix Corp., Hinesburg, VT, USA);2 RGB cameras C525 (Logitech, Lausanne, Switzerland);2 environmental sensors DAS43X (Holland Scientific, Inc., Lincoln, NE, USA);1 GNSS unit 19X (Garmin Ltd., Olathe, KS, USA).

In addition to the previously mentioned sensors, other components in this subsystem included a laptop computer as the control terminal, two 8-Port USB to Serial Hubs (StarTech.com Ltd., London, ON, Canada), and two power banks of 20,000 mAh working as batteries. The power banks, serial hubs, and their connectors were located inside plastic enclosures for additional protection. The computer ran the Graphical User Interface (GUI) for the operator to monitor and adjust the behavior of the system. The serial hubs are required to facilitate the connections of the previously mentioned sensors to the terminal. Figure 2 illustrates the connections between the different electronic components of the HTPP. In particular, M was 2 and N was 3 in the built platform, but the versatility of the design allows the number of sensor blocks to be reduced or increased as required by the size, budget, and nature of the experiment.

While the handheld setup used the older model of the multispectral sensor ACS-430, the vehicle-mounted HTPP platform used the newer model ACS-435. Both models measure reflectance at 3 bands: NIR (780 nm), Red-Edge (730 nm), and Red (670 nm), as well as reporting two Vegetation Indexes (VI): NDVI and NDRE:(1)$$NDVI=\frac{\rho_{NIR}-\rho_{Red}}{\rho_{NIR}+\rho_{Red}}$$
(2)$$NDRE=\frac{\rho_{NIR}-\rho_{RedEdge}}{\rho_{NIR}+\rho_{RedEdge}}$$

For both setups, an additional VI was computed by the custom software: Chlorophyll Index (CI), as defined by:(3)$$CI=\frac{\rho_{NIR}}{\rho_{RedEdge}}-1$$

While other indices might be applicable, NDVI was chosen because it is the VI with the most widespread usage. NDVI is a good biomass estimator and reveals aggregating numbers of the size and density of the canopy [33]. NDRE is a similar measure that is used to overcome problems with saturation at late phenological stages; this is one of the shortcomings of NDVI. In other words, the expectation was that NDVI would perform better during early crop stages, while NDRE would become more relevant at later stages [34]. For example, NDRE has been found to perform better than NDVI in predicting SPAD-based chlorophyll in quinoa [35]. While both NDVI and NDRE are popular for their general-purpose applicability, CI was chosen as a more focused VI to complement the other two, since it correlates with the total chlorophyll content of the leaves [36].

The ultrasonic sensor measured distance between itself and the target canopy, which then could be used to compute plant height, since the height of the sensor holder is predefined. Both the ultrasonic and multispectral sensors provide information about crop status. The height measured from the ultrasonic sensor is an important parameter of crop architectonics. Some breeding techniques have been used to adjust the height of certain crops to better match the range covered by combine harvesters. Height can be related to canopy volume. The different VIs measured are not used as frequently as traits by themselves, but they can relate to multiple other properties such as plant vigor and canopy coverage. The combination of both types of sensors is central to the concept of the sensing units. There are many advantages to the fusion of the two types of measurements. Among them is the possibility of separating plant growth from plant nutrition when studying measurement trends. For example, they could be used to improve the prediction of biomass, where it could be assumed that height relates to volume of the canopy and the VIs with its density. Thus, other sensors may provide similar types of information related to canopy size and stress. Hyperspectral sensors and LiDAR could be combined for the same reasons. However, multispectral and ultrasonic measurements were chosen for this platform due to their lower cost, higher reliability, and simplicity in operation and data processing.

The properties measured by the environmental sensor DAS43X were air temperature, air humidity, atmospheric pressure, incident Photosynthetically Active Radiation (PAR), and reflected PAR. Incorporating these measurements is particularly important to guarantee the transferability of models built using the system, and to perform multi-year experiments.

The cameras were included for real-time feedback during the operation of the platform. They could potentially be used to increase information value, but this was outside the scope of the current work.

### 2.3. Mechanical Subsystem of the HTPP Platform

For the vehicle-mounted setup, the main structural component of the mechanical subsystem is a bar clamp with two swivel pads on each jaw pivot to grip nearly any shape. This provides confidence in the mounting capabilities of the brackets on different types of vehicles or sprayer booms without prior knowledge of their exact geometry. Sensing units can be mounted on each side of the sprayer boom or a similar toolbar using aluminum L-brackets bolted to the bar clamp. Ease of installation was a major consideration in designing the system. The mechanical design was made using Autodesk Inventor (Autodesk, Inc., San Rafael, CA, USA).

The vehicle, a Gator 850D XUV (John Deere, Moline, IL, USA), was equipped with a custom horizontal beam, made by mounting aluminum bars with steel L-brackets to its trunk, so they could be extended 1.5 m to the sides of the vehicle. A steel thread connected to the furthermost ends of the beam was used to keep the beam bars straight. A small winch was used to tighten the steel thread. 3D printed brackets were made to attach cameras to the top corners of the frame of the Gator, as well as the GNSS receiver. Figure 3 shows an image of the system. During testing, it was verified that two operators could install and uninstall the sensors with ease, with no additional tools required.

### 2.4. Software Subsystem of HTPP Platform

A Python script relying on open-source libraries was made to log the measurements from all sensors into text files and display the GUI. Within it, there is information for the operator about the connected sensors, a map tracking the trajectory of the vehicle, video streaming of the cameras, and real-time plots of the sensor readings. A bar of tabs was used to change between the plots of the several measured variables. Certain settings could be modified, such as the spacing between the sensors, which offered the ability to disengage any subset of sensors. Figure 4 shows a screenshot of the main window.

### 2.5. Experimental Design

#### 2.5.1. Experimental Design of Benchmark Experiment with Handheld Setup

An initial benchmark experiment took place during the summer of 2019, when the handheld setup was used. The plants were sown on 21 June 2019, at the Emile A. Lods Agronomy Research Center of Macdonald Campus, McGill University, divided into 22 plots, arranged in a grid of 2 × 11. The 11 different bean varieties/cultivars (*Phaseolus vulgaris* L.) were Apex, Argosy, Calmant, Compass, Dresden, Knight rider, Majesty, Mast, Nautica, Red rider, and Sheek. All varieties had two replicates. In each plot, there were four crop rows. Measurements were taken on each plot on the dates listed in Table 1, including their geo-references and timestamps. For each plot and date, eight locations were selected to increase the variety of sampling sites for data collection. All plots were exposed to the same management practices and were of the same size (6.75 m^2^). The plants were harvested on 7 October 2019.

#### 2.5.2. Experimental Design of Experiment with Vehicle-Mounted Platform

A second experiment was performed during the summer of 2020, where the vehicle-mounted HTPP platform was used. This time, plants were sown on 1 June 2020, divided among 242 plots, arranged in a grid of 11 × 22. This experiment took place in a different field of the Emile A. Lods Agronomy Research Center of Macdonald Campus, McGill University. Eleven different bean varieties were grown and considered for the data analysis: Rampart, Blackbeard, Etna, Apex, T9905, Windbreaker, Zorro, Merlot, Nautica, Redhawk, and Red Rover. Two different factors of management practices were varied: row spacing of either 22 in. (559 mm) or 30 in. (762 mm), and the plant density 10,000 plants/acre (~24,700 plants/ha) or 15,000 plants/acre (37,100 plants/ha), respectively. Again, all plots were the same size (~15 m^2^) and included four crop rows. The timing of this experiment was affected by COVID-19 and its related restrictions. Table 2 illustrates the dates of data collection that were used for the analysis. On these dates, the vehicle-mounted HTPP platform would drive through the alleys in between plots as close to a constant speed of 0.8 m/s as possible and collect measurements on-the-go from each side. A new sensor reading was processed every 6 ms. Cameras were set to record at about 3 fps. The plants were harvested on 6 October 2020. Soil sampling was conducted on 23–24 October 2020 at six locations around the field. The soil cores collected during the sampling were divided into shallow (between surface level and 152 mm) and deep (between 152 mm and 305 mm), stored in soil bags, and labelled for laboratory analysis. The laboratory analysis was performed on 29 October 2020 by A & L Labs (A & L Canada Laboratories Inc., London, ON, Canada) to measure the soil chemical properties such as the concentration of N, P, and K, among others. Figure 5 summarizes all the variables involved in the experiments. The public records of a nearby weather station operated by Environment and Climate Change Canada at coordinates 45°25′38″ N, 73°55′45″ W (about 700 m away from the location of both fields) were referenced.

### 2.6. Data Analysis

#### 2.6.1. Data Analysis of Benchmark Experiment with Handheld Setup

For the benchmark experiment with the handheld setup, the raw data were saved as text files, which were parsed to comma-separated values file format using a Python script. In the same script, the longitude and latitude values reported by the GNSS receiver were projected into planar coordinates. The phenotypical variables (NDVI, NDRE, CI, plant height) were then grouped by plot. A simple average of each plot was computed. This process was repeated separately for each of the dates where data were collected.

The results per plot and date were then imported into MATLAB (The MathWorks, Inc., Natick, MA, USA) for additional processing, comprised of computing repeated measurements ANOVA for each of the phenotypical variables as dependent variables, where crop varieties and date were the independent variables.

#### 2.6.2. Data Analysis of Experiment with Vehicle-Mounted Platform

Just as for the benchmark experiment with the handheld setup, for the experiment with the vehicle-mounted HTPP platform the raw data were saved as text files, which were parsed to comma-separated values file format using a Python script. In the same script, the longitude and latitude values reported by the GNSS receiver were projected into planar coordinates. The phenotypical variables (NDVI, NDRE, CI, plant height) and the environmental factors (air temperature, air moisture, incident PAR) were then grouped by plot based on geographical location. The maximum and the average of values above a threshold were calculated per plot. The threshold was defined using the histograms of the readings, which showed a bimodal distribution, and the principle of Otsu’s method [37] was used to find the optimal threshold that maximizes inter-class variance. The histograms with a bimodal distribution were assumed to be the result of the sum of two normal distributions: one for measurements of soil and another for measurements of plant tissue, as exemplified by Figure 6. This process was repeated separately for each of the dates where data were collected. These values were used to populate maps for each variable and date. The results per plot and date were then imported into MATLAB (The MathWorks, Inc., Natick, MA, USA) for additional processing.

The HTPP platform described here could serve as a tool for on-farm experimentation with different goals. Thus, the present data analysis procedure illustrates a possible approach using the available data. In addition to the creation of the previously mentioned maps, which showcase the spatial variability of the phenotypical traits, the data analysis comprised computing:Histograms of the phenotypical variables for each date to visualize the temporal variability across the population;Averages and standard deviations of the phenotypical variables per crop variety and date;Built linear models; where the phenotypical data for each of the measurement dates plus the yield were the dependent variables; and the management practices, soil, and measured weather factors, as shown in Figure 5, were independent variables.

Each linear model was iteratively evaluated by adding new variables, one at a time out of the set of predictors. The coefficient of determination was evaluated for each date and then averaged across all dates. The models which maximized the average coefficient of determination across all six dates were chosen to prevent overfitting for specific dates. The soil measurements were interpolated using Inverse Distance Weighting (IDW). The weather variables were considered in four ways:Only the weather data of the same date as the phenotypical data were used;Only the weather data of the previous measured date as the phenotypical data were used;The average of all weather data until the same date as the phenotypical data was used;The average of all weather data until the previous measured date as the phenotypical data was used.

For the first date of measurements, since there are no previous days, only method (a) was used. Once the best models were found, the residuals were computed by subtracting the predicted value from the actual value. Then a one-way ANOVA was performed with variety as the explanatory factor. Tukey’s test was performed to evaluate the effect of the varieties on the residuals. This process was carried out to illustrate an expected pipeline that uses the data provided by the platform and how its capabilities can be exploited.

## 3. Results and Discussion

### 3.1. Benchmark Handheld Experiment

The results of the ANOVA with repeated measurements over time are shown in Figure 7. This test shows that both time, and the interaction between the variety and time, are significant for all evaluated VIs as well as plant height, given that all the *p*-values are below 0.05, even when the Greenhouse-Geisser adjustment is applied (pValueGG in the Figure 7). This means that the effect of the variety as a discrete factor varies with time, when considering its explanatory power in the measured plant traits.

### 3.2. Vehicle-Mounted Experiment

Figure 8 illustrates the locations of each plot in the field, highlighting those of the Apex, Windbreaker, and Redhawk varieties, which will be used as examples throughout this section. Specifically, these three varieties were selected to represent the varieties with high, medium, and low yield, respectively. Figure 9, Figure 10 and Figure 11 present some of the properties mapped by the system. As these are the output of the system, they indicate the ability of the platform to complete its task. For a few plots, no phenotypical data were available. There were two possible reasons for this: (1) the plants sown there were removed at an earlier date or (2) they were inaccessible to the HTPP platform because of the presence of a fence in the field. In these maps, the measurements from the DAS43X sensor were aggregated by averaging over the plots. For the multispectral and ultrasonic sensor, it was important to reduce the effect of soil patches which were measured inside the field of view of the sensor, given the spacing between the crop rows. Because of this, the phenotypical data were aggregated per plot by two methods: (1) finding the maximum and (2) calculating the mean of the values above a certain threshold, as explained in Section 2.5.

Finally, Figure 12 displays the average behavior of the variables across time for the example varieties, along with the standard deviation. In general, the values of the measured variables decreased as time passed, as was expected given the overall trend in the first experiment and the phenological stage of the crop around the dates covered by the experiment.

Table 3 recounts the variables which were selected in the iterative process to build the linear models to provide the best fit, following the procedure described earlier. For the weather factors, the number in between brackets identifies the approach used to account for the fact that the variables had been measured multiple times. The first approach, which used only the information from the same date as the target phenotypical variable, performed the best in most cases. For certain dates, the previous measurement date provided an even better prediction, but, as mentioned previously, only the best performing option across all dates was selected. Only in the case of height was it found that some of the other approaches worked best for all dates. Shallow organic matter was found in all models, confirming it as a variable that consistently has explanatory power. The same can be said for air temperature and PAR, which are key atmospheric variables, included in all the built models. Although these results are valuable, it is the procedure used to obtain this type of information that must be highlighted as a tool for future experiments in the field.

Table 4 compares the effect sizes of the groups of explanatory variables for CI and height for the second date (20 August). The component labelled as Error could be further reduced by the inclusion of some of the properties not directly considered, such as soil moisture, which was excluded in this analysis as it was not included in the standard laboratory analysis. While interaction factors among the current variables were evaluated, none were found to be significant in this dataset; therefore, they were removed. In general, the trend was that the error diminished with time, as it was possible to explain a larger portion of the variability in the plant traits through the evaluated factors. However, it was accompanied by a less significant portion of the total effect size due to the variety component.

### 3.3. Comparison

The average duration of the data collection for the first experiment with the handheld setup was slightly more than 1.5 h, for a field size of 150 m^2^. The rate was 100 m^2^/h. With the vehicle-mounted HTPP platform, the average time was a little more than 45 min, for a field of size 3600 m^2^. Consequently, the rate was 5400 m^2^/h, more than 50 times faster, even with the low travel speed of the vehicle as it scanned the field. In the context of breeding, this could represent an improvement in genetic gain of the order of 50.

The vehicle-mounted platform has the advantage of being able to include both environmental sensors and cameras, making it more ergonomic for the operator who does not have to carry the sensors. This is increased by the modularity and adaptability of the system. However, the handheld setup has the benefit of allowing the operator to safely reach inner parts of the plots, which is too risky for the vehicle. The 8 measured locations in a plot, which represent about 10% of the plot’s area, are distributed around all possible surfaces and may be a more representative sample of the population. In contrast, the vehicle-mounted platform covers almost 40% of the plot’s surface, but it is concentrated near the edges of the plot closest to the alleys where driving is possible. While the vehicle-mounted platform provides more consistent distancing between the sensors and the target, it comes at the cost of increased vibrations from the vehicle’s engine. Mounting the system on a high-clearance vehicle would provide access to more areas of the field, as is the trend in similar scenarios. An interesting possibility is to mount the system on an autonomous vehicle that can monitor the field continuously.

## 4. Conclusions

The proposed vehicle-mounted HTPP platform is capable of mapping phenotypical and environmental data across a field, as evidenced by Figure 9, Figure 10 and Figure 11. The system’s capability to measure multispectral crop data and merge them with ultrasonic and environmental parameters is key. Furthermore, as supported by the discussion in Section 3.3, it can be argued that the HTPP platform fulfills this function in an efficient manner, even when compared to the handheld setup; it is evident that both options have strengths and drawbacks. The different statistical tests which were evaluated illustrate multiple approaches to screen for crop varieties in terms of the evolution of their phenotypical traits over time. Crop modelling techniques benefit from the availability of such data at multiple scales. More complex techniques could benefit from the general workflow discussed herewith, enabled by the availability of data gathered by the platform. The versatility of the platform facilitates its adaptation to different fields and locations, including the ability to be extended with additional sensors. Holistic analysis allows for models to be built to serve as tools to evaluate relative importance of different factors of plant growth. The central role of organic matter, air temperature, and incident PAR in the plant processes is evidenced by their relevance in the feature selection step of the model building. Between 50% and 75% of the variability in the phenotypical variables was explained by the linear models, including 6 to 8 explanatory variables related to the environmental, managerial, and genetic factors of plant growth. Further research with more varieties of crops and in different geographical locations is required to evaluate this framework.

Determining if the investment in a HTPP platform is justified for a given grower will ultimately depend on many factors, such as field size and the goal of the grower. Nonetheless, the quick-install approach, the use of open-source software for the GUI interfacing with sensors, the modular design of the sensor units and their brackets, the selection of accessible types of sensors, the inclusion of envirotyping on the mobile platform complemented by the weather station data, and the aggregation method inspired by Otsu’s method are concepts that should be explored in developing future HTPP platforms. Additional adaptations would serve to achieve more widespread adoption of HTPP platforms for outdoor environments. Further research is required to leverage the video recordings; this is a potential resource which was underutilized in this work. This research work is being continued with a focus on using the aforementioned tools to study the variability at the subplot level.

## Figures and Tables

**Figure 1 sensors-23-04253-f001:**
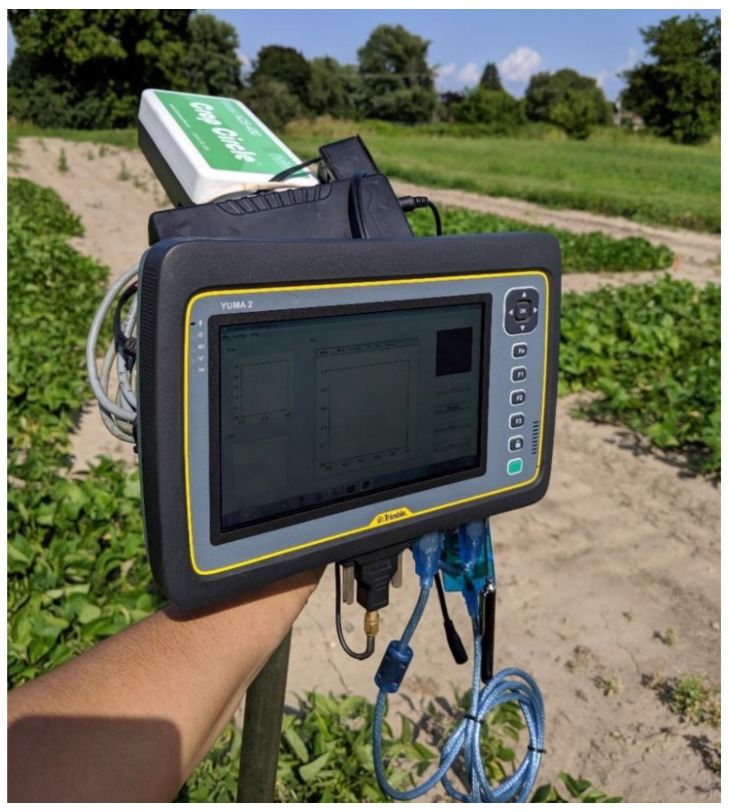
Picture of handheld setup.

**Figure 2 sensors-23-04253-f002:**
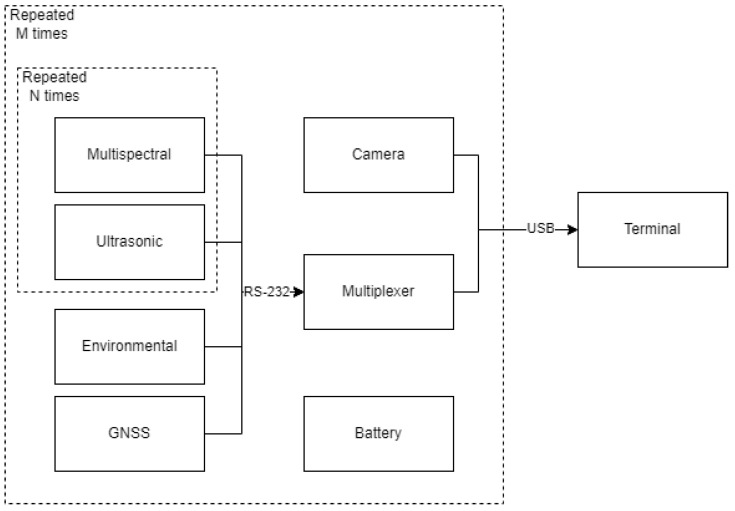
Block diagram of the electronic subsystem of the HTPP platform.

**Figure 3 sensors-23-04253-f003:**
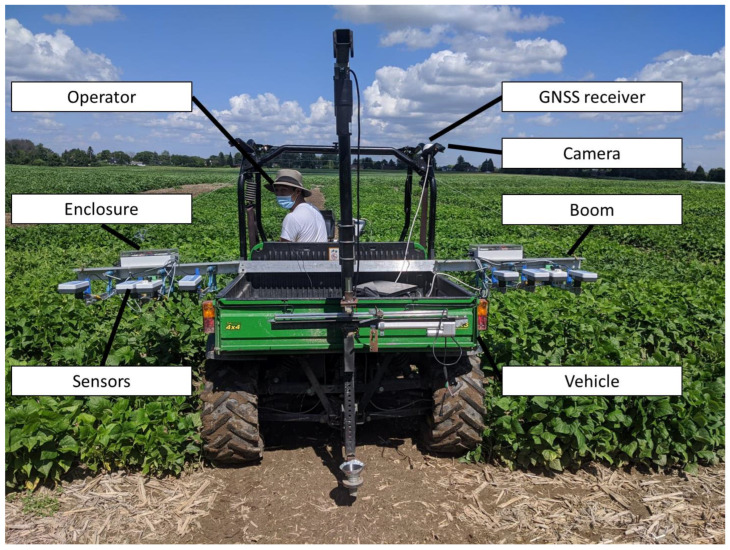
Picture of built prototype of vehicle-mounted HTPP setup.

**Figure 4 sensors-23-04253-f004:**
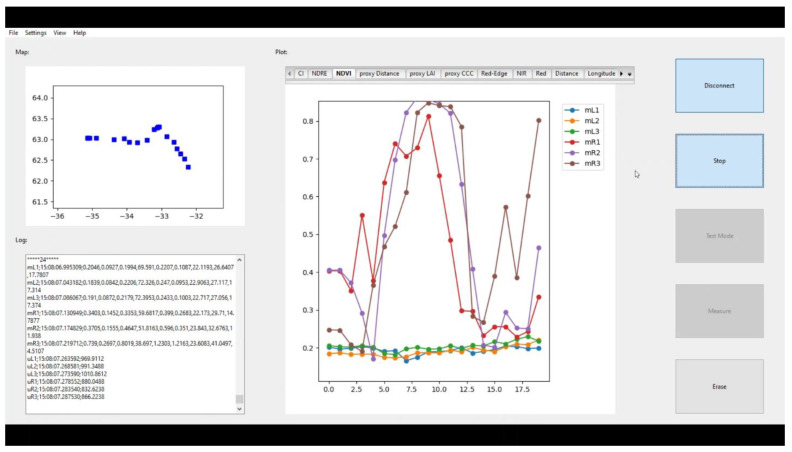
Screenshot of GUI (main window).

**Figure 5 sensors-23-04253-f005:**
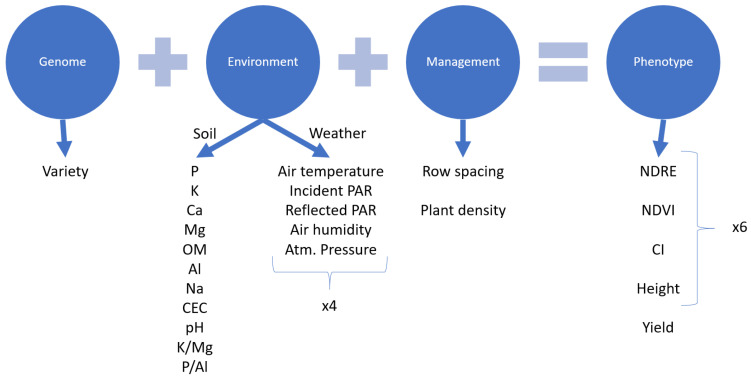
Diagram of variables collected in experiments and used to build models.

**Figure 6 sensors-23-04253-f006:**
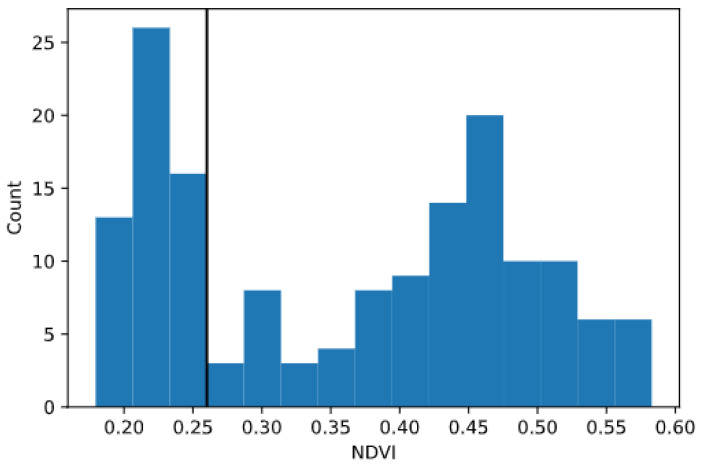
Example histogram of NDVI at last date for a plot and estimated threshold that separates bimodal distribution.

**Figure 7 sensors-23-04253-f007:**
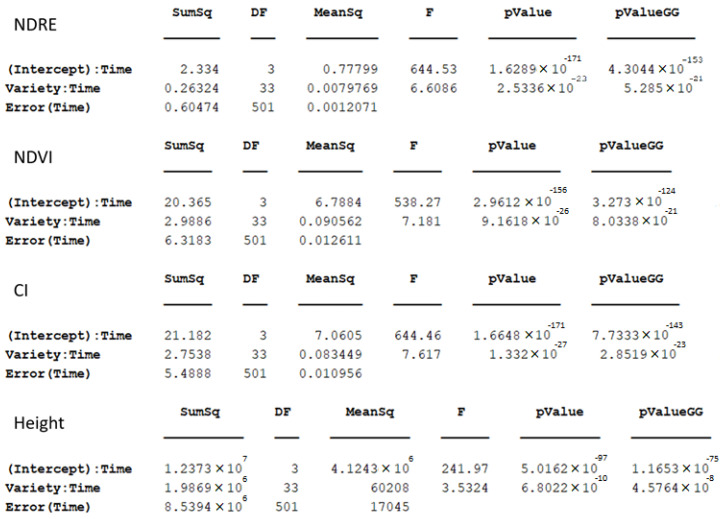
Results of repeated measurements ANOVA over time.

**Figure 8 sensors-23-04253-f008:**
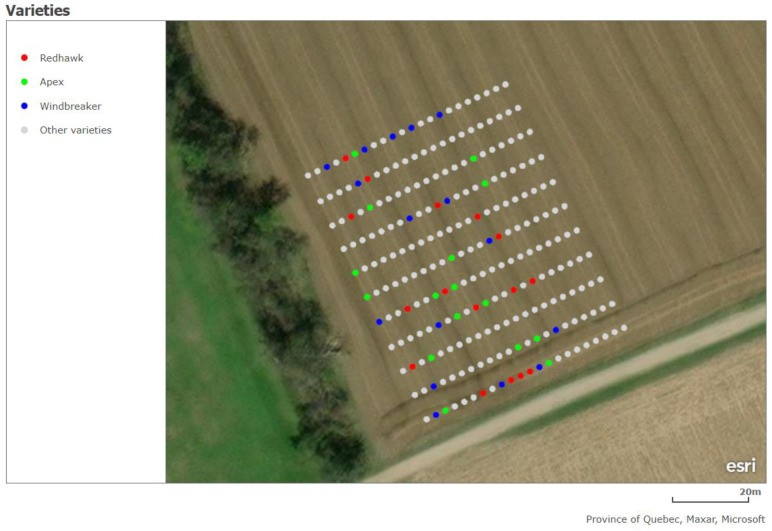
Map of plot locations highlighting plots of example varieties.

**Figure 9 sensors-23-04253-f009:**
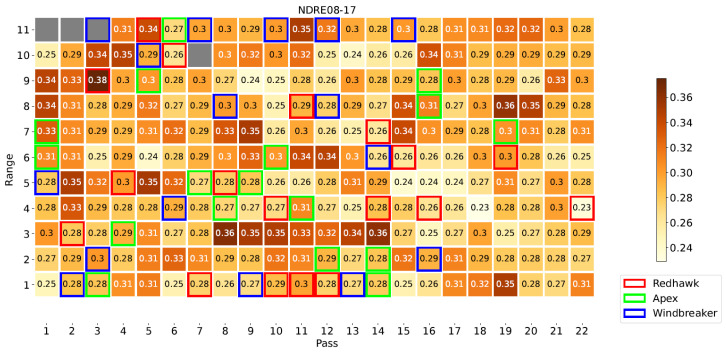
Map of NDRE at first date using maximum for aggregation highlighting example varieties.

**Figure 10 sensors-23-04253-f010:**
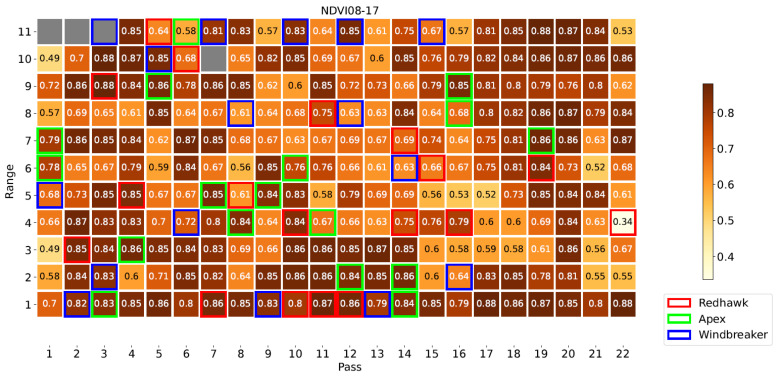
Map of NDVI at first date using average for aggregation highlighting example varieties.

**Figure 11 sensors-23-04253-f011:**
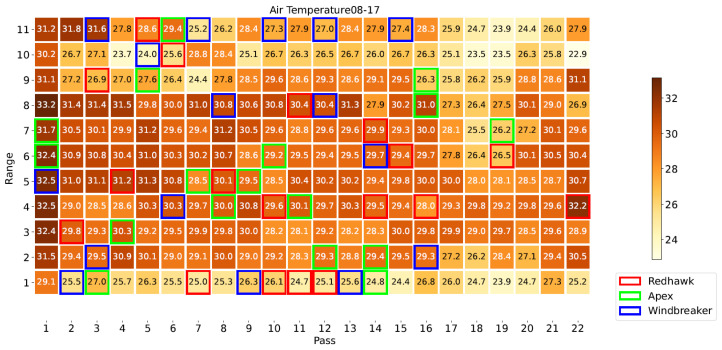
Map of air temperature in °C at first date highlighting example varieties.

**Figure 12 sensors-23-04253-f012:**
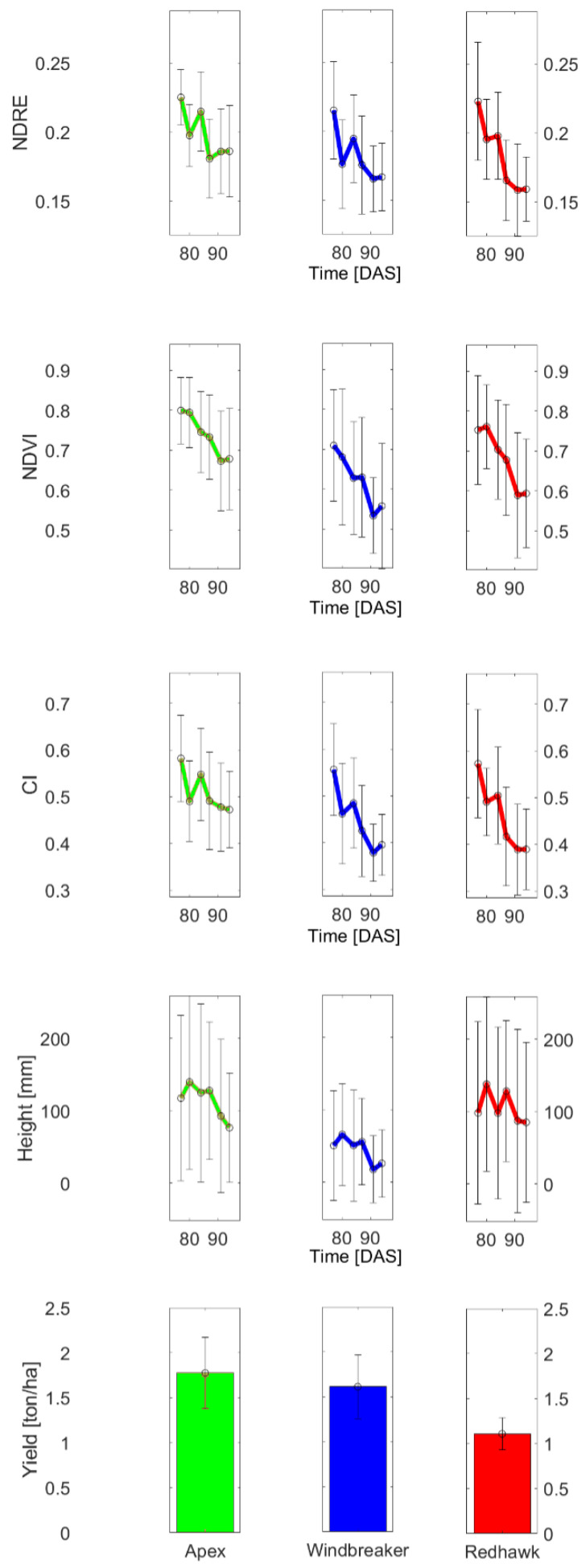
Evolution of phenotypical data for example varieties.

**Table 1 sensors-23-04253-t001:** Dates of data collection for first experiment.

Date	Days after Seeding	Start Time	End Time
5 July 2019	14	12:31	14:12
25 July 2019	34	16:07	17:32
20 August 2019	60	14:20	16:05
18 September 2019	89	13:03	14:43

**Table 2 sensors-23-04253-t002:** Dates of data collection for second experiment.

Date	Days after Seeding	Start Time	End Time
17 August 2020	77	13:04	13:49
20 August 2020	80	15:47	16:28
24 August 2020	84	12:49	13:29
27 August 2020	87	14:27	15:10
31 August 2020	91	13:32	14:13
3 September 2020	94	15:35	16:15

**Table 3 sensors-23-04253-t003:** Variables selected for linear models of each phenotypical trait that maximized coefficient of determination across all dates.

Dependent Variable	Independent Variables
NDRE	Shallow organic matter
	Deep Na
	Air temperature [1]
	Incident PAR [1]
	Reflected PAR [1]
	Variety
NDVI	Shallow organic matter
	Deep Na
	Deep Mg
	Air temperature [1]
	Incident PAR [1]
	Reflected PAR [1]
	Variety
CI	Shallow organic matter
	Deep Na
	Deep Mg
	Air temperature [1]
	Incident PAR [1]
	Reflected PAR [1]
	Row spacing
	Variety
Height	Shallow organic matter
	Deep Ca
	Air temperature [3]
	Incident PAR [1]
	Reflected PAR [2]
	Row spacing
	Variety
Yield	Shallow organic matter
	Shallow Ca
	Air temperature [1]
	Incident PAR [1]
	Reflected PAR [1]
	Row spacing
	Variety

**Table 4 sensors-23-04253-t004:** Effect sizes by category of explanatory variables on CI and height for 20 August.

Category	Effect Sizes on CI [%]	Effect Sizes on Height [%]
Environment	34	69
Management	5	1
Variety	8	4
Error	54	25

## Data Availability

Not applicable.

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
