# Peer review of "Development of a Quick-Install Rapid Phenotyping System"

_sensors, 2023, doi:10.3390/s23094253_

Round 1

Reviewer 1 Report

This article presents a cost-effective phenotyping system and real data collected by this system. The work has a huge potencial, however the article has several serious problems and lacks in terms of clarity. It share a lot of data that are no easy to read and understand. You should focus in bring more information and less data.  

It is not clear where the authors wants to reach with this work. What is the biggest contribution? this should be clear in the abstract. Is the solution? or is the analyse and knowledge extraction from the data collected? 

In the abstract is mentioned "Comparing the results of this system against a manual setup show a throughput increase with a factor of 50, supporting the notion that such a system allows for mapping crop status across a field in an efficient manner.". From this statement is expected the validation of the phenotyping system, but what we get is the analyse of data collected, no benchmark is realized against other solutions.  

If the focus is the system, you should benchmark the system against a ground truth system, this is not presented on the article.

If the focus is the phenotyping data, it would be benefical that the data is public  and focus on this open dataset analyses. Otherwise the contribution of this article for the community is very short. 

Other issues:

The acronyms are not consistente presented: 

line 39- Light detection and ranging (LiDAR)..... should Light Detection And Ranging (LiDAR)

you present also: Wireless Sensor Networks (WSN) and QTLs (Quantitative Trait Loci), please make this uniform. VI is visual index?

Fig. 2 should be improved to make readable and understandable. 

how ACS-435 can extract the LAI? is not clear this is possible. 

why CI is measured like presented in eq 3? why this correct and why you choose this way, there are other ways, and there is no reference to sustain the eq. 3. 

line 176 is mentioned "NDVI was chosen because it is the VI with the most widespread usage." this does not necessarily means that is the correct way to do it and extract meaningful information. 

section 2.2

It is mentioned two vehicles, but in the end is used Gator 850D XUV, what is the relation between these two vehicles? 

fig3 needs to be improved in terms quality and readability

in 306 is GPS or GNSS receiver?

408-410 not clear the text. I cannot understand the idea-  

Fig. 7 and 8 is the plant density or prescription map? no clear, seems to be just density map . If is the prescription map, what was applied and in what units ?

fig. 18 to 23 a lot of data without no meaningful information. The full article presents a lot of data that is hard to read and understand. Please keep the focus in bring information and not just data.

Reviewer 2 Report

Authors presents a sperimentation related to a "quick install" sensor system for phenotyping.

In my opinion this paper is  interesting, well organized, and results presented are complete. My olny suggestion to improve the quality of this work is:

- extend a bit more the state of the art analysis (if not a dedicated section, extend the introduction)

- provide the reader with some more details about the specific scenario indicators, in order to make the paper more "self contained" and  understandable for all readers. In example NDVI and NDRE indicators can be introduced with some more details, and also selected sensors can be introduced with some more details. 
